# Importance Weighting and Variational Inference

**Justin Domke**[1] **and Daniel Sheldon**[1,2]
[1] College of Information and Computer Sciences, University of Massachusetts Amherst
[2] Department of Computer Science, Mount Holyoke College

## Abstract

Recent work used importance sampling ideas for better variational bounds on likelihoods. We clarify the applicability of these ideas to pure probabilistic inference, by showing the resulting Importance Weighted Variational Inference (IWVI) technique is an instance of augmented variational inference, thus identifying the looseness in previous work. Experiments confirm IWVI's practicality for probabilistic inference. As a second contribution, we investigate inference with elliptical distributions, which improves accuracy in low dimensions, and convergence in high dimensions.

## 1 Introduction

Probabilistic modeling is used to reason about the world by formulating a joint model $p(\mathbf{z}, \mathbf{x})$ for unobserved variables $\mathbf{z}$ and observed variables $\mathbf{x}$, and then querying the posterior distribution $p(\mathbf{z} \mid \mathbf{x})$ to learn about hidden quantities given evidence $\mathbf{x}$. Common tasks are to draw samples from $p(\mathbf{z} \mid \mathbf{x})$ or compute posterior expectations. However, it is often intractable to perform these tasks directly, so considerable research has been devoted to methods for approximate probabilistic inference.

Variational inference (VI) is a leading approach for approximate inference. In VI, $p(\mathbf{z} \mid \mathbf{x})$ is approximated by a distribution $q(\mathbf{z})$ in a simpler family for which inference is tractable. The process to select $q$ is based on the following decomposition [22, Eqs. 11-12]:

$$\log p(\mathbf{x}) = \underbrace{\mathop{\mathbb{E}}_{q(\mathbf{z})} \log \frac{p(\mathbf{z}, \mathbf{x})}{q(\mathbf{z})}}_{\text{ELBO}[q(\mathbf{z}) \| p(\mathbf{z},\mathbf{x})]} + \underbrace{\text{KL}\left[q(\mathbf{z}) \| p(\mathbf{z}|\mathbf{x})\right]}_{\text{divergence}}. \tag{1}$$

The first term is a lower bound of $\log p(\mathbf{x})$ known as the "evidence lower bound" (ELBO). Selecting $q$ to make the ELBO as big as possible simultaneously obtains a lower bound of $\log p(\mathbf{x})$ that is as tight as possible and drives $q$ close to $p$ in KL-divergence.

The ELBO is closely related to importance sampling. For fixed $q$, let $R = p(\mathbf{z}, \mathbf{x})/q(\mathbf{z})$ where $\mathbf{z} \sim q$. This random variable satisfies $p(\mathbf{x}) = \mathbb{E} R$, which is the foundation of importance sampling. Similarly, we can write by Jensen's inequality that $\log p(\mathbf{x}) \geq \mathbb{E} \log R = \text{ELBO}\,[q\|p]$, which is the foundation of modern "black-box" versions of VI (BBVI) [19] in which Monte Carlo samples are used to estimate $\mathbb{E} \log R$, in the same way that IS estimates $\mathbb{E} R$.

Critically, the *only* property VI uses to obtain a lower bound is $p(\mathbf{x}) = \mathbb{E} R$. Further, it is straightforward to see that Jensen's inequality yields a tighter bound when $R$ is more concentrated about its mean $p(\mathbf{x})$. So, it is natural to consider different random variables with the same mean that are more concentrated, for example the sample average $R_M = \frac{1}{M} \sum_{m=1}^{M} R_m$. Then, by identical reasoning, $\log p(\mathbf{x}) \geq \mathbb{E} \log R_M$. The last quantity is the objective of importance-weighted auto-encoders [5]; we call it the *importance weighted ELBO (IW-ELBO)*, and the process of selecting $q$ to maximize it *importance-weighted VI (IWVI)*.

However, at this point we should pause. The decomposition in Eq. 1 makes it clear exactly in what sense standard VI, when optimizing the ELBO, makes $q$ close to $p$. By switching to the one-dimensional random variable $R_M$, we derived the IW-ELBO, which gives a tighter bound on $\log p(\mathbf{x})$. For learning applications, this may be all we want. But for probabilistic inference, we are left uncertain exactly in what sense $q$ "is close to" $p$, and how we should use $q$ to approximate $p$, say, for computing posterior expectations.

Our first contribution is to provide a new perspective on IWVI by highlighting a precise connection between IWVI and *self-normalized importance sampling* (NIS) [17], which instructs us how to use IWVI for "pure inference" applications. Specifically, IWVI is an instance of augmented VI. Maximizing the IW-ELBO corresponds exactly to minimizing the KL divergence between joint distributions $q_M$ and $p_M$, where $q_M$ is derived from NIS over a batch of $M$ samples from $q$, and $p_M$ is the joint distribution obtained by drawing one sample from $p$ and $M - 1$ "dummy" samples from $q$. This has strong implications for probabilistic inference (as opposed to learning) which is our primary focus. After optimizing $q$, one should compute posterior expectations using NIS. We show that not only does IWVI significantly tighten bounds on $\log p(\mathbf{x})$, but, by using $q$ this way at test time, it significantly reduces estimation error for posterior expectations.

Previous work has connected IWVI and NIS by showing that the importance weighted ELBO is a lower bound of the ELBO applied to the NIS distribution [6, 16, 2]. Our work makes this relationship precise as an instance of augmented VI, and exactly quantifies the gap between the IW-ELBO and conventional ELBO applied to the NIS distribution, which is a conditional KL divergence.

Our second contribution is to further explore the connection between variational inference and importance sampling by adapting ideas of "defensive sampling" [17] to VI. Defensive importance sampling uses a widely dispersed $q$ distribution to reduce variance by avoiding situations where $q$ places essentially no mass in an area with $p$ has density. This idea is incompatible with regular VI due to its "mode seeking" behavior, but it is quite compatible with IWVI. We show how to use elliptical distributions and reparameterization to achieve a form of defensive sampling with almost no additional overhead to black-box VI (BBVI). "Elliptical VI" provides small improvements over Gaussian BBVI in terms of ELBO and posterior expectations. In higher dimensions, these improvements diminish, but elliptical VI provides significant improvement in the convergence reliability and speed. This is consistent with the notion that using a "defensive" $q$ distribution is advisable when it is not well matched to $p$ (e.g., before optimization has completed).

## 2  Variational Inference

Consider again the "ELBO decomposition" in Eq. 1. Variational inference maximizes the "evidence lower bound" (ELBO) over $q$. Since the divergence is non-negative, this tightens a lower-bound on $\log p(\mathbf{x})$. But, of course, since the divergence and ELBO vary by a constant, maximizing the ELBO is equivalent to minimizing the divergence. Thus, variational inference can be thought of as simultaneously solving two problems:

- **"probabilistic inference"** or finding a distribution $q(\mathbf{z})$ that is close to $p(\mathbf{z}|\mathbf{x})$ in KL-divergence.

- **"bounding the marginal likelihood"** or finding a lower-bound on $\log p(\mathbf{x})$.

The first problem is typically used with Bayesian inference: A user specifies a model $p(\mathbf{z}, \mathbf{x})$, observes some data $\mathbf{x}$, and is interested in the posterior $p(\mathbf{z}|\mathbf{x})$ over the latent variables. While Markov chain Monte Carlo is most commonly for these problems [9, 23], the high computational expense motivates VI [11, 3]. While a user might be interested in any aspect of the posterior, for concreteness, we focus on "posterior expectations", where the user specifies some arbitrary $t(\mathbf{z})$ and wants to approximate $\mathbb{E}_{p(\mathbf{z}|\mathbf{x})} t(\mathbf{z})$.

The second problem is typically used to support maximum likelihood learning. Suppose that $p_\theta(\mathbf{z}, \mathbf{x})$ is some distribution over observed data $\mathbf{x}$ and hidden variables $\mathbf{z}$. In principle, one would like to set $\theta$ to maximize the marginal likelihood over the observed data. When the integral $p_\theta(\mathbf{x}) = \int p_\theta(\mathbf{z}, \mathbf{x}) d\mathbf{z}$ is intractable, one can optimize the lower-bound $\mathbb{E}_{q(\mathbf{z})} \log \left( p_\theta(\mathbf{z}, \mathbf{x})/q(\mathbf{z}) \right)$ instead [22], over both $\theta$ and the parameters of $q$. This idea has been used to great success recently with variational auto-encoders (VAEs) [10].

## 3 Importance Weighting

Recently, ideas from importance sampling have been applied to obtain tighter ELBOs for learning in VAEs [5]. We review the idea and then draw novel connections to augmented VI that make it clear how adapt apply these ideas to probabilistic inference.

Take any random variable $R$ such that $\mathbb{E}\, R = p(\mathbf{x})$, which we will think of as an "estimator" of $p(\mathbf{x})$. Then it's easy to see via Jensen's inequality that

$$\log p(\mathbf{x}) = \underbrace{\mathbb{E}\log R}_{\text{bound}} + \underbrace{\mathbb{E}\log \frac{p(\mathbf{x})}{R}}_{\text{looseness}}, \quad (2)$$

where the first term is a lower bound on $\log p(\mathbf{x})$, and the second (non-negative) term is the looseness. The bound will be tight if $R$ is highly concentrated.

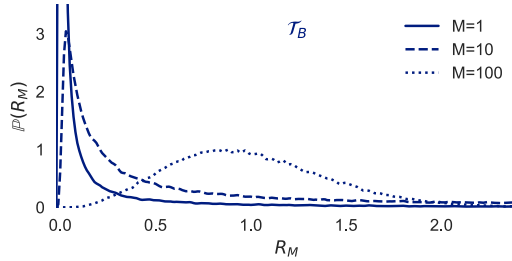

Figure 1: How the density of $R_M$ changes with $M$. (Distribution and setting as in Fig. 2.)

While Eq. 2 looks quite trivial, it is a generalization of the "ELBO" decomposition in Eq. 1. To see that, use the random variable

$$R = \omega(\mathbf{z}) = \frac{p(\mathbf{z}, \mathbf{x})}{q(\mathbf{z})},\ \mathbf{z} \sim q, \quad (3)$$

which clearly obeys $\mathbb{E}\, R = p(\mathbf{x})$, and for which Eq. 2 becomes Eq. 1.

The advantage of Eq. 2 over Eq. 1 is increased flexibility: alternative estimators $R$ can give a tighter bound on $\log p(\mathbf{x})$. One natural idea is to draw multiple i.i.d. samples from $q$ and average the estimates as in importance sampling (IS) . This gives the estimator

$$R_M = \frac{1}{M}\sum_{m=1}^{M} \frac{p\left(\mathbf{z}_m, \mathbf{x}\right)}{q(\mathbf{z}_m)},\ \mathbf{z}_m \sim q. \quad (4)$$

It's always true that $\mathbb{E}\, R_M = p(\mathbf{x})$, but the distribution of $R_M$ places less mass near zero for larger $M$, which leads to a tighter bound (Fig. 1).

This leads to a tighter "importance weighted ELBO" (IW-ELBO) lower bound on $\log p(\mathbf{x})$, namely

$$\text{IW-ELBO}_M\left[q(\mathbf{z})\|p(\mathbf{z}, \mathbf{x})\right] := \underset{q(\mathbf{z}_{1:M})}{\mathbb{E}}\log \frac{1}{M}\sum_{m=1}^{M} \frac{p\left(\mathbf{z}_m, \mathbf{x}\right)}{q(\mathbf{z}_m)}, \quad (5)$$

where $\mathbf{z}_{1:M}$ is a shorthand for $(\mathbf{z}_1, ..., \mathbf{z}_M)$ and $q(\mathbf{z}_{1:M}) = q(\mathbf{z}_1)\cdots q(\mathbf{z}_M)$. This bound was first introduced by Burda et al. [5] in the context of supporting maximum likelihood learning of a variational auto-encoder.

### 3.1 A generative process for the importance weighted ELBO

While Eq. 2 makes clear that optimizing the IW-ELBO tightens a bound on $\log p(\mathbf{x})$, it isn't obvious what connection this has to probabilistic inference. Is there some divergence that is being minimized? Theorem 1 shows this can be understood by constructing "augmented" distributions $p_M(\mathbf{z}_{1:M}, \mathbf{x})$ and $q_M(\mathbf{z}_{1:M})$ and then applying the ELBO decomposition in Eq. 1 to the joint distributions.

**Theorem 1** (IWVI). *Let $q_M(\mathbf{z}_{1:M})$ be the density of the generative process described by Alg. 1, which is based on self-normalized importance sampling over a batch of $M$ samples from $q$. Let $p_M\left(\mathbf{z}_{1:M}, \mathbf{x}\right) = p(\mathbf{z}_1, \mathbf{x})q(\mathbf{z}_{2:M})$ be the density obtained by drawing $\mathbf{z}_1$ and $\mathbf{x}$ from $p$ and drawing the "dummy" samples $\mathbf{z}_{2:M}$ from $q$. Then*

$$q_M\left(\mathbf{z}_{1:M}\right) = \frac{p_M(\mathbf{z}_{1:M}, \mathbf{x})}{\frac{1}{M}\sum_{m=1}^{M}\omega(\mathbf{z}_m)}. \quad (6)$$

*Further, the ELBO decomposition in Eq. 1 applied to $q_M$ and $p_M$ is*

$$\log p(\mathbf{x}) = \text{IW-ELBO}_M\left[q(\mathbf{z})\|p(\mathbf{z}, \mathbf{x})\right] + \text{KL}\left[q_M(\mathbf{z}_{1:M})\|p_M(\mathbf{z}_{1:M}|\mathbf{x})\right]. \quad (7)$$

---

**Algorithm 1** A generative process for $q_M(\mathbf{z}_{1:M})$

---

1. Draw $\hat{\mathbf{z}}_1, \hat{\mathbf{z}}_1, ..., \hat{\mathbf{z}}_M$ independently from $q(\mathbf{z})$.

2. Choose $m \in \{1, ..., M\}$ with probability $\dfrac{\omega(\hat{\mathbf{z}}_m)}{\sum_{m'=1}^{M} \omega(\hat{\mathbf{z}}_{m'})}$.

3. Set $\mathbf{z}_1 = \hat{\mathbf{z}}_m$ and $\mathbf{z}_{2:M} = \hat{\mathbf{z}}_{-m}$ and return $\mathbf{z}_{1:M}$.

---

We will call the process of maximizing the IW-ELBO "Importance Weighted Variational Inference" (IWVI). (Burda et al. used "Importance Weighted Auto-encoder" for optimizing Eq. 5 as a bound on the likelihood of a variational auto-encoder, but this terminology ties the idea to a particular model, and is not suggestive of the probabilistic inference setting.)

The generative process for $q_M$ in Alg. 1 is very similar to self-normalized importance sampling. The usual NIS distribution draws a batch of size $M$, and then "selects" a single variable with probability in proportion to its importance weight. NIS is exactly equivalent to the marginal distribution $q_M(\mathbf{z}_1)$. The generative process for $q_M(\mathbf{z}_{1:M})$ additionally keeps the *unselected* variables and relabels them as $\mathbf{z}_{2:M}$.

Previous work [6, 2, 16, 12] investigated a similar connection between NIS and the importance-weighted ELBO. In our notation, they showed that

$$\log p(\mathbf{x}) \geq \text{ELBO}\left[q_M(\mathbf{z}_1) \| p(\mathbf{z}_1, \mathbf{x})\right] \geq \text{IW-ELBO}_M\left[q(\mathbf{z}) \| p(\mathbf{z}, \mathbf{x})\right]. \tag{8}$$

That is, they showed that the IW-ELBO *lower bounds* the ELBO between the NIS distribution and $p$, without quantifying the gap in the second inequality. Our result makes it clear exactly what KL-divergence is being minimized by maximizing the IW-ELBO and in what sense doing this makes $q$ "close to" $p$. As a corollary, we also quantify the gap in the inequality above (see Thm. 2 below).

A recent decomposition [12, Claim 1] is related to Thm. 1, but based on different augmented distributions $q_M^{IS}$ and $p_M^{IS}$. This result is fundamentally different in that it holds $q_M^{IS}$ "fixed" to be an independent sample of size $M$ from $q$, and modifies $p_M^{IS}$ so its marginals approach $q$. This does not inform inference. Contrast this with our result, where $q_M(\mathbf{z}_1)$ gets closer and closer to $p(\mathbf{z}_1 \mid \mathbf{x})$, and can be used for probabilistic inference. See appendix (Section A.3.2) for details.

Identifying the precise generative process is useful if IWVI will be used for general probabilistic queries, which is a focus of our work, and, to our knowledge, has not been investigated before. For example, the expected value of $t(\mathbf{z})$ can be approximated as

$$\mathbb{E}_{p(\mathbf{z}|\mathbf{x})} t(\mathbf{z}) = \mathbb{E}_{p_M(\mathbf{z}_1|\mathbf{x})} t(\mathbf{z}_1) \approx \mathbb{E}_{q_M(\mathbf{z}_1)} t(\mathbf{z}_1) = \mathbb{E}_{q(\mathbf{z}_{1:M})} \frac{\sum_{m=1}^{M} \omega(\mathbf{z}_m)\, t(\mathbf{z}_m)}{\sum_{m=1}^{M} \omega(\mathbf{z}_m)}. \tag{9}$$

The final equality is established by Lemma 4 in the Appendix. Here, the inner approximation is justified since IWVI minimizes the joint divergence between $q_M(\mathbf{z}_{1:M})$ and $p_M(\mathbf{z}_{1:M}|\mathbf{x})$. However, this is *not* equivalent to minimizing the divergence between $q_M(\mathbf{z}_1)$ and $p_M(\mathbf{z}_1|\mathbf{x})$, as the following result shows.

**Theorem 2.** *The marginal and joint divergences relevant to IWVI are related by*

$$\text{KL}\left[q_M(\mathbf{z}_{1:M}) \| p_M(\mathbf{z}_{1:M}|\mathbf{x})\right] = \text{KL}\left[q_M(\mathbf{z}_1) \| p(\mathbf{z}_1|\mathbf{x})\right] + \text{KL}\left[q_M(\mathbf{z}_{2:M}|\mathbf{z}_1) \| q(\mathbf{z}_{2:M})\right].$$

*As a consequence, the gap in the first inequality of Eq. 8 is exactly* $\text{KL}\left[q_M(\mathbf{z}_1) \| p(\mathbf{z}_1|\mathbf{x})\right]$ *and the gap in the second inequality is exactly* $\text{KL}\left[q_M(\mathbf{z}_{2:M}|\mathbf{z}_1) \| q(\mathbf{z}_{2:M})\right]$.

The first term is the divergence between the marginal of $q_M$, i.e., the "standard" NIS distribution, and the posterior. In principle, this is exactly the divergence we would like to minimize to justify Eq. 9. However, the second term is not zero since the selection phase in Alg. 1 leaves $\mathbf{z}_{2:M}$ distributed differently under $q_M$ than under $q$. Since this term is irrelevant to the quality of the approximation in Eq. 9, IWVI truly minimizes an upper-bound. Thus, IWVI can be seen as an instance of auxiliary variational inference [1] where a joint divergence upper-bounds the divergence of interest.

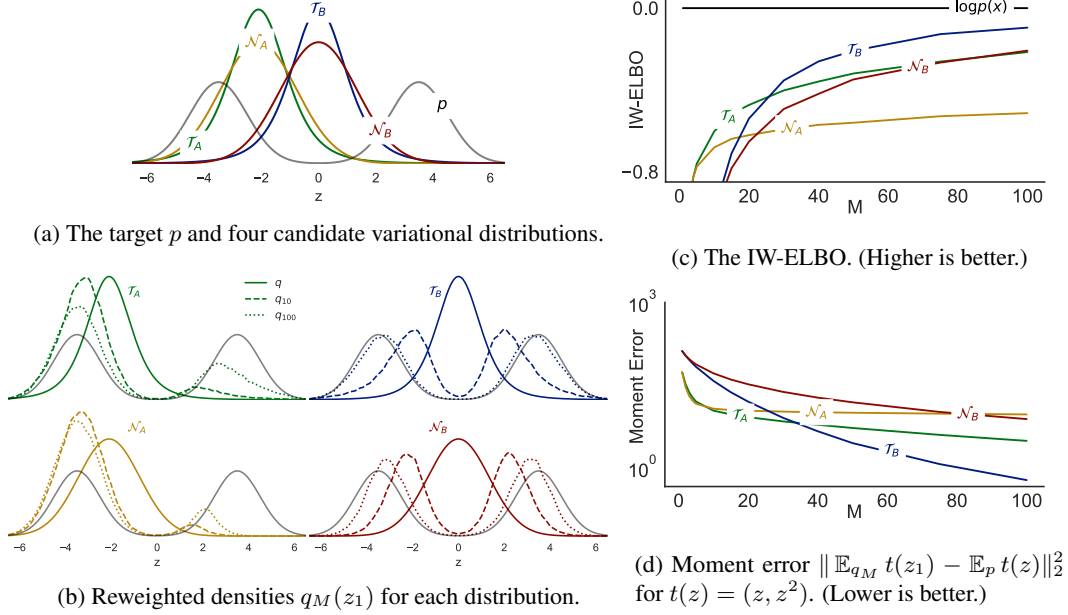

(a) The target $p$ and four candidate variational distributions.

(c) The IW-ELBO. (Higher is better.)

(b) Reweighted densities $q_M(z_1)$ for each distribution.

(d) Moment error $\| \mathbb{E}_{q_M} t(z_1) - \mathbb{E}_p t(z) \|_2^2$ for $t(z) = (z, z^2)$. (Lower is better.)

Figure 2: Two Gaussian ($\mathcal{N}$) and two Student-T ($\mathcal{T}$) variational distributions, all with constant variance and one of two means ($A$ or $B$). For $M = 1$ it is better to use a mean closer to one mode of $p$. For large $M$, a mean in the center is superior, and the heavy tails of the Student T lead to better approximation of $p$ and better performance both in terms of IW-ELBO and moment error.

## 4    Importance Sampling Variance

This section considers the family for the variational distribution. For small $M$, the mode-seeking behavior of VI will favor weak tails, while for large $M$, variance reduction provided by importance weighting will favor wider tails.

The most common variational distribution is the Gaussian. One explanation for this is the Bayesian central limit theorem, which, in many cases, guarantees that the posterior is asymptotically Gaussian. Another is that it's "safest" to have weak tails: since the objective is $\mathbb{E} \log R$, small values of $R$ are most harmful. So, VI wants to avoid cases where $q(\mathbf{z}) \gg p(\mathbf{z}, \mathbf{x})$, which is difficult if $q$ is heavy-tailed. (This is the "mode-seeking" behavior of the KL-divergence [24].)

With IWVI, the situation changes. Asymptotically in $M$, $R_M$ in Eq. 4 concentrates around $p(\mathbf{x})$, and so it is the variance of $R_M$ that matters, as formalized in the following result.

**Theorem 3.** *For large $M$, the looseness of the IW-ELBO is given by the variance of $R$. Formally, if there exists some $\alpha > 0$ such that $\mathbb{E} |R - p(\mathbf{x})|^{2+\alpha} < \infty$ and $\limsup_{M \to \infty} \mathbb{E}[1/R_M] < \infty$, then*

$$\lim_{M \to \infty} M \underbrace{\left( \log p(\mathbf{x}) - \text{IW-ELBO}_M \left[ q(\mathbf{z}) \| p(\mathbf{z}, \mathbf{x}) \right] \right)}_{\text{KL}[q_M \| p_M]} = \frac{\mathbb{V}[R]}{2 p(\mathbf{x})^2}.$$

Maddison et al. [13] give a related result. Their Proposition 1 applied to $R_M$ gives the same conclusion (after an argument based on the Marcinkiewicz-Zygmund inequality; see appendix) but requires the sixth central moment to exist, whereas we require only existence of $\mathbb{E} |R - p(\mathbf{x})|^{2+\alpha}$ for any $\alpha > 0$. The $\limsup$ assumption on $\mathbb{E} 1/R_M$ is implied by assuming that $\mathbb{E} 1/R_M < \infty$ for any finite $M$ (or for $R$ itself). Rainforth et al. [18, Theorem 1 in Appendix] provide a related asymptotic for errors in gradient variance, assuming at least the third moment exists.

Directly minimizing the variance of $R$ is equivalent to minimizing the $\chi^2$ divergence between $q(\mathbf{z})$ and $p(\mathbf{z}|\mathbf{x})$, as explored by Dieng et al. [7]. Overdispersed VI [21] reduces the variance of score-function estimators using heavy-tailed distributions.

The quantity inside the parentheses on the left-hand side is exactly the KL-divergence between $q_M$ and $p_M$ in Eq. 7, and accordingly, even for constant $q$, this divergence asymptotically decreases at a $1/M$ rate.

The variance of $R$ is a well-explored topic in traditional importance sampling. Here the situation is reversed from traditional VI– since $R$ is non-negative, it is very *large* values of $R$ that can cause high variance, which occurs when $q(\mathbf{z}) \ll p(\mathbf{z}, \mathbf{x})$. The typical recommendation is "defensive sampling" or using a widely-dispersed proposal [17]. For these reasons, we believe that the best form for $q$ will vary depending on the value of $M$. Figure 1 explores a simple example of this in 1-D.

## 5 Elliptical Distributions

Elliptical distributions are a generalization of Gaussians that includes the Student-T, Cauchy, scale-mixtures of Gaussians, and many others. The following short review assumes a density function exists, enabling a simpler presentation than the typical one based on characteristic functions [8].

We first describe the special case of spherical distributions. Take some density $\rho(r)$ for a non-negative $r$ with $\int_0^\infty \rho(r) = 1$. Define the **spherical random variable** $\boldsymbol{\epsilon}$ corresponding to $\rho$ as

$$\boldsymbol{\epsilon} = r\mathbf{u}, \; r \sim \rho, \; \mathbf{u} \sim S, \tag{10}$$

where $S$ represents the uniform distribution over the unit sphere in $d$ dimensions. The density of $\boldsymbol{\epsilon}$ can be found using two observations. First, it is constant for all $\boldsymbol{\epsilon}$ with a fixed radius $\|\boldsymbol{\epsilon}\|$. Second, if if $q_{\boldsymbol{\epsilon}}(\boldsymbol{\epsilon})$ is integrated over $\{\boldsymbol{\epsilon} : \|\boldsymbol{\epsilon}\| = r\}$ the result must be $\rho(r)$. Using these, it is not hard to show that the density must be

$$q_{\boldsymbol{\epsilon}}(\boldsymbol{\epsilon}) = g(\|\boldsymbol{\epsilon}\|_2^2), \quad g(a) = \frac{1}{S_{d-1}a^{(d-1)/2}}\rho\left(\sqrt{a}\right), \tag{11}$$

where $S_{d-1}$ is the surface area of the unit sphere in $d$ dimensions (and so $S_{d-1}a^{(d-1)/2}$ is the surface area of the sphere with radius $a$) and $g$ is the **density generator**.

Generalizing, this, take some positive definite matrix $\Sigma$ and some vector $\mu$. Define the **elliptical random variable z** corresponding to $\rho, \Sigma$, and $\boldsymbol{\mu}$ by

$$\mathbf{z} = rA^\top\mathbf{u} + \boldsymbol{\mu}, \; r \sim \rho, \; \mathbf{u} \sim S, \tag{12}$$

where $A$ is some matrix such that $A^\top A = \Sigma$. Since $\mathbf{z}$ is an affine transformation of $\boldsymbol{\epsilon}$, it is not hard to show by the "Jacobian determinant" formula for changes of variables that the density of $\mathbf{z}$ is

$$q(\mathbf{z}|\boldsymbol{\mu}, \Sigma) = \frac{1}{|\Sigma|^{1/2}}g\left((\mathbf{z} - \boldsymbol{\mu})^T\Sigma^{-1}(\mathbf{z} - \boldsymbol{\mu})\right), \tag{13}$$

where $g$ is again as in Eq. 11. The mean and covariance are $\mathbb{E}[\mathbf{z}] = \boldsymbol{\mu}$, and $\mathbb{C}[\mathbf{z}] = \left(\mathbb{E}[r^2]/d\right)\Sigma$.

For some distributions, $\rho(r)$ can be found from observing that $r$ has the same distribution as $\|\boldsymbol{\epsilon}\|$. For example, with a Gaussian, $r^2 = \|\boldsymbol{\epsilon}\|^2$ is a sum of $d$ i.i.d. squared Gaussian variables, and so, by definition, $r \sim \chi_d$.

## 6 Reparameterization and Elliptical Distributions

Suppose the variational family $q(\mathbf{z}|w)$ has parameters $w$ to optimize during inference. The reparameterization trick is based on finding some density $q_{\boldsymbol{\epsilon}}(\boldsymbol{\epsilon})$ independent of $w$ and a "reparameterization function" $\mathcal{T}(\boldsymbol{\epsilon}; w)$ such that $\mathcal{T}(\boldsymbol{\epsilon}; w)$ is distributed as $q(\mathbf{z}|w)$. Then, the ELBO can be re-written as

$$\text{ELBO}[q(\mathbf{z}|w)\|p(\mathbf{z}, \mathbf{x})] = \mathop{\mathbb{E}}_{q_{\boldsymbol{\epsilon}}(\boldsymbol{\epsilon})} \log \frac{p(\mathcal{T}(\boldsymbol{\epsilon}; w), \mathbf{x})}{q(\mathcal{T}(\boldsymbol{\epsilon}; w)|w)}.$$

The advantage of this formulation is that the expectation is independent of $w$. Thus, computing the gradient of the term inside the expectation for a random $\boldsymbol{\epsilon}$ gives an unbiased estimate of the gradient. By far the most common case is the multivariate Gaussian distribution, in which case the base density $q_{\boldsymbol{\epsilon}}(\boldsymbol{\epsilon})$ is just a standard Gaussian and for some $A_w$ such that $A_w^\top A_w = \Sigma_w$,

$$\mathcal{T}(\boldsymbol{\epsilon}; w) = A_w^\top\boldsymbol{\epsilon} + \boldsymbol{\mu}_w. \tag{14}$$

## 6.1 Elliptical Reparameterization

To understand Gaussian reparameterization from the perspective of elliptical distributions, note the similarity of Eq. 14 to Eq. 12. Essentially, the reparameterization in Eq. 14 combines $r$ and $\mathbf{u}$ into $\epsilon = r\mathbf{u}$. This same idea can be applied more broadly: for any elliptical distribution, *provided the density generator $g$ is independent of $w$*, the reparameterization in Eq. 14 will be valid, provided that $\epsilon$ comes from the corresponding spherical distribution.

While this independence is true for Gaussians, this is not the case for other elliptical distributions. If $\rho_w$ itself is a function of $w$, Eq. 14 must be generalized. In that case, think of the generative process (for $v$ sampled uniformly from $[0, 1]$)

$$\mathcal{T}(\mathbf{u}, v; w) = F_w^{-1}(v) A_w^T \mathbf{u} + \boldsymbol{\mu}_w, \tag{15}$$

where $F_w^{-1}(v)$ is the inverse CDF corresponding to the distribution $\rho_w(r)$. Here, we should think of the vector $(\mathbf{u}, v)$ playing the role of $\epsilon$ above, and the base density as $q_{\mathbf{u}, v}(\mathbf{u}, v)$ being a spherical density for $\mathbf{u}$ and a uniform density for $v$.

To calculate derivatives with respect to $w$, backpropagation through $A_w$ and $\boldsymbol{\mu}_w$ is simple using any modern autodiff system. So, if the inverse CDF $F_w^{-1}$ has a closed-form, autodiff can be directly applied to Eq. 15. If the inverse CDF does not have a simple closed-form, the following section shows that only the CDF is actually needed, provided that one can at least sample from $\rho(r)$.

## 6.2 Dealing CDFs without closed-form inverses

For many distributions $\rho$, the inverse CDF may not have a simple closed form, yet highly efficient samplers still exist (most commonly custom rejection samplers with very high acceptance rates). In such cases, one can still achieve the *effect* of Eq. 15 on a random $v$ using only the CDF (not the inverse). The idea is to first directly generate $r \sim \rho_w$ using the specialized sampler, and only then find the corresponding $v = F_w(r)$ using the closed-form CDF. To understand this, observe that if $r \sim \rho$ and $v \sim \text{Uniform}[0, 1]$, then the pairs $(r, F_w(r))$ and $(F_w^{-1}(v), v)$ are identically distributed. Then, via the implicit function theorem, $\nabla_w F_w^{-1}(v) = -\nabla_w F_w(r) / \nabla_r F_w(r)$. All gradients can then be computed by "pretending" that one had started with $v$ and computed $r$ using the inverse CDF.

## 6.3 Student T distributions

The following experiments will consider student T distributions. The spherical T distribution can be defined as $\epsilon = \sqrt{\nu} \boldsymbol{\delta} / s$ where $\boldsymbol{\delta} \sim \mathcal{N}(0, I)$ and $s \sim \chi_\nu$ [8]. Equivalently, write $r = \|\epsilon\| = \sqrt{\nu} t / s$ with $t \sim \chi_d$. This shows that $r$ is the ratio of two independent $\chi$ variables, and thus determined by an F-distribution, the CDF of which could be used directly in Eq. 15. We found a slightly "bespoke" simplification helpful. As there is no need for gradients with respect to $d$ (which is fixed), we represent $\epsilon$ as $\epsilon = (\sqrt{\nu} t / s) \mathbf{u}$, leading to reparameterizing the elliptical T distribution as

$$\mathcal{T}(\mathbf{u}, t, v; w) = \frac{\sqrt{\nu} t}{F_\nu^{-1}(v)} A_w^\top \mathbf{u} + \boldsymbol{\mu}_w,$$

where $F_\nu$ is the CDF for the $\chi_\nu$ distribution. This is convenient since the CDF of the $\chi$ distribution is more widely available than that of the F distribution.

# 7 Experiments

All the following experiments compare "E-IWVI" using student T distributions to "IWVI" using Gaussians. Regular "VI" is equivalent to IWVI with $M = 1$.

We consider experiments on three distributions. In the first two, a computable $\log p(\mathbf{x})$ enables estimation of the KL-divergence and computable true mean and variance of the posterior enable a precise evaluation of test integral estimation. On these, we used a fixed set of $10,000 \times M$ random inputs to $\mathcal{T}$ and optimized using batch L-BFGS, avoiding heuristic tuning of a learning rate sequence.

A first experiment considered random Dirichlet distributions $p(\boldsymbol{\theta}|\boldsymbol{\alpha})$ over the probability simplex in $K$ dimensions, $\boldsymbol{\theta} \in \Delta^K$. Each parameter $\alpha_k$ is drawn i.i.d. from a Gamma distribution with a shape parameter of 10. Since this density is defined only over the probability simplex, we borrow from Stan

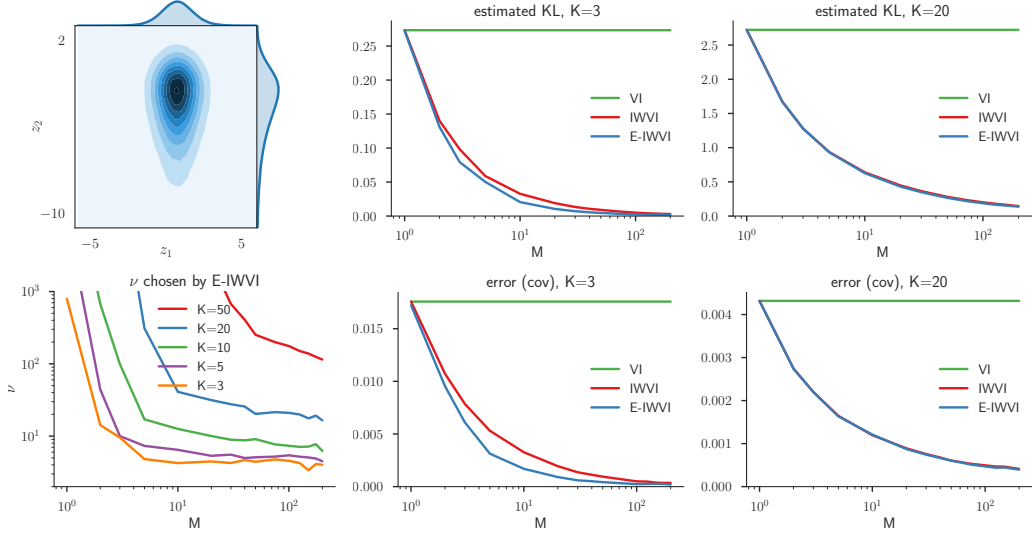

Figure 3: Random Dirichlets, averaged over 20 repetitions. Top left shows an example posterior for $K = 3$. The test-integral error is $\|\mathbb{C}[\boldsymbol{\theta}] - \hat{\mathbb{C}}[\boldsymbol{\theta}]\|_F$ where $\hat{\mathbb{C}}$ is the empirical covariance of samples drawn from $q_M(\mathbf{z}_1)$ and then transformed to $\Delta^K$. In all cases, IWVI is able to reduce the error of VI to negligible levels. E-IWVI provides an accuracy benefit in low dimensions but little when $K = 20$.

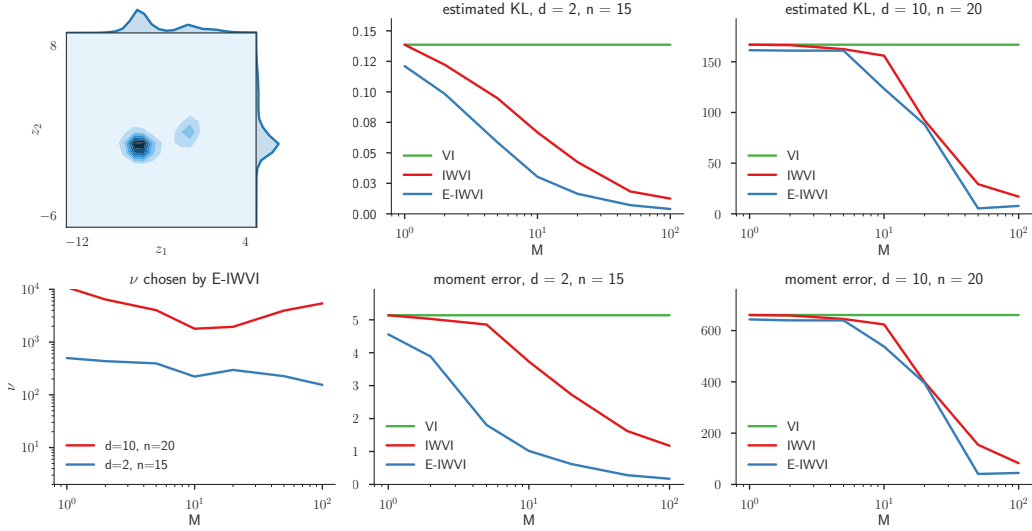

Figure 4: Clutter Distributions, averaged over 50 repetitions. The error shows the error in the estimated second moment $\mathbb{E}[\mathbf{z}\mathbf{z}^T]$. IWVI reduces the errors of VI by orders of magnitude. E-IWVI provides a diminishing benefit in higher dimensions.

the strategy of transforming to an unconstrained $\mathbf{z} \in \mathbb{R}^{K-1}$ space via a stick-breaking process [23]. To compute test integrals over variational distributions, the reverse transformation is used. Results are shown in Fig. 3.

A second experiment uses Minka's "clutter" model [15]: $\mathbf{z} \in \mathbb{R}^d$ is a hidden object location, and $\mathbf{x} = (\mathbf{x}_1, \ldots, \mathbf{x}_n)$ is a set of $n$ noisy observations, with $p(\mathbf{z}) = \mathcal{N}(\mathbf{z}; \mathbf{0}, 100I)$ and $p(\mathbf{x}_i | \mathbf{z}) = 0.25\mathcal{N}(\mathbf{x}_i; \mathbf{z}, I) + 0.75\mathcal{N}(\mathbf{x}_i; 0, 10I)$. The posterior $p(\mathbf{z} \mid \mathbf{x})$ is a mixture of $2^n$ Gaussians, for which we can do exact inference for moderate $n$. Results are shown in Fig. 4.

Finally, we considered a (non-conjugate) logistic regression model with a Cauchy prior with a scale of 10, using stochastic gradient descent with various step sizes. On these higher dimensional problems, we found that when the step-size was perfectly tuned and optimization had many iterations, both methods performed similarly in terms of the IW-ELBO. E-IWVI never performed worse, and

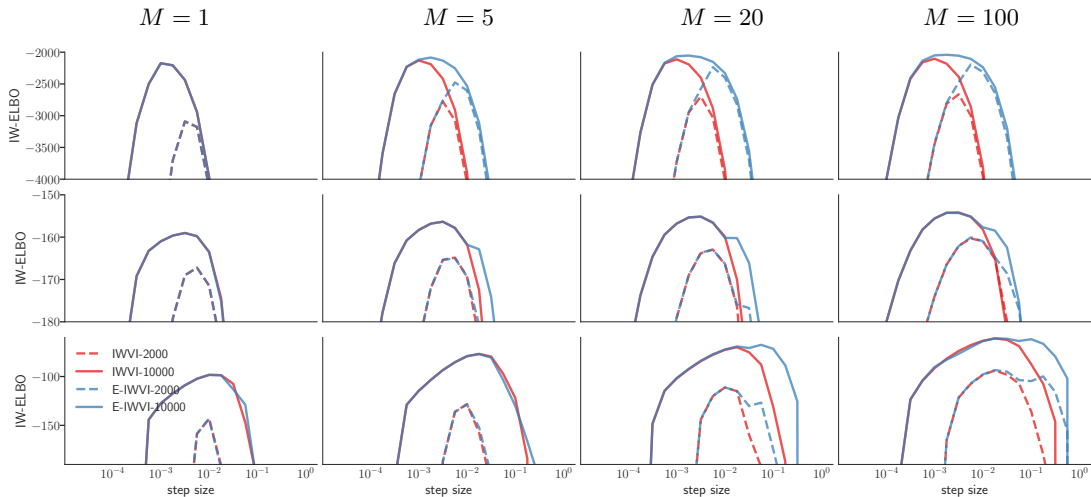

Figure 5: Logistic regression comparing IWVI (red) and E-IWVI (blue) with various $M$ and step sizes. The IW-ELBO is shown after 2,000 (dashed lines) and 10,000 (solid) iterations. A larger $M$ consistently improves both methods. E-IWVI converges more reliably, particularly on higher-dimensional data. From top: Madelon ($d = 500$) Sonar ($d = 60$), Mushrooms ($d = 112$).

sometimes performed very slightly better. E-IWVI exhibited superior convergence behavior and was easier to tune, as illustrated in Fig. 5, where E-IWVI converges at least as well as IWVI for *all* step sizes. We suspect this is because when $w$ is far from optimal, both the IW-ELBO and gradient variance is better with E-IWVI.

### Acknowledgements

We thank Tom Rainforth for insightful comments regarding asymptotics and Theorem 3 and Linda Siew Li Tan for comments regarding Lemma 7. This material is based upon work supported by the National Science Foundation under Grant No. 1617533.

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
