[Supplementary Material]

# A Appendix

## A.1 Additional Experimental Results

Figure 6: More figures corresponding to the 1-D example.

Figure 7: More Results on Inference with Dirichlet distributions

Figure 8: Logistic regression experiments (as in Fig. 5) on more datasets.

## A.2  Proofs for Section 3

**Theorem 1** (IWVI). *Let $q_M(\mathbf{z}_{1:M})$ be the density of the generative process described by Alg. 1, which is based on self-normalized importance sampling over a batch of $M$ samples from $q$. Let $p_M(\mathbf{z}_{1:M}, \mathbf{x}) = p(\mathbf{z}_1, \mathbf{x})q(\mathbf{z}_{2:M})$ be the density obtained by drawing $\mathbf{z}_1$ and $\mathbf{x}$ from $p$ and drawing the "dummy" samples $\mathbf{z}_{2:M}$ from $q$. Then*

$$q_M(\mathbf{z}_{1:M}) = \frac{p_M(\mathbf{z}_{1:M}, \mathbf{x})}{\frac{1}{M}\sum_{m=1}^{M}\omega(\mathbf{z}_m)}. \tag{6}$$

*Further, the ELBO decomposition in Eq. 1 applied to $q_M$ and $p_M$ is*

$$\log p(\mathbf{x}) = \text{IW-ELBO}_M\left[q(\mathbf{z})\|p(\mathbf{z},\mathbf{x})\right] + \text{KL}\left[q_M(\mathbf{z}_{1:M})\|p_M(\mathbf{z}_{1:M}|\mathbf{x})\right]. \tag{7}$$

*Proof.* For the density $q_M$, define the distribution

$$
\begin{aligned}
q_M(\hat{\mathbf{z}}_{1:M}, \mathbf{z}_{1:M}, h) &= q_M(\hat{\mathbf{z}}_{1:M})\, q_M(h|\hat{\mathbf{z}}_{1:M})\, q_M(\mathbf{z}_{1:M}|\hat{\mathbf{z}}_{1:M}, h) \\
q_M(\hat{\mathbf{z}}_{1:M}) &= q(\hat{\mathbf{z}}_{1:M}) \\
q_M(h|\hat{\mathbf{z}}_{1:M}) &= \frac{p(\hat{\mathbf{z}}_h, x)/q(\hat{\mathbf{z}}_h)}{\sum_{m=1}^{M} p(\hat{\mathbf{z}}_m, x)/q(\hat{\mathbf{z}}_m)} \\
q_M(\mathbf{z}_{1:M}|\hat{\mathbf{z}}_{1:M}, h) &= \delta(\mathbf{z}_1 - \hat{\mathbf{z}}_h)\,\delta(\mathbf{z}_{2:M} - \hat{\mathbf{z}}_{-h}).
\end{aligned}
$$

What is the marginal distribution over $\mathbf{z}_{1:M}$?

$$
\begin{aligned}
q_M(\mathbf{z}_{1:M}) &= \int \sum_{h=1}^{M} q_M(\hat{\mathbf{z}}_{1:M})\, q_M(h|\hat{\mathbf{z}}_{1:M})\, q_M(\mathbf{z}_{1:M}|\hat{\mathbf{z}}_{1:M}, h)\, d\hat{\mathbf{z}}_{1:M} \\
&= \int \sum_{h=1}^{M} q(\hat{\mathbf{z}}_{1:M}) \frac{p(\hat{\mathbf{z}}_h, \mathbf{x})/q(\hat{\mathbf{z}}_h)}{\sum_{m=1}^{M} p(\hat{\mathbf{z}}_m, \mathbf{x})/q(\hat{\mathbf{z}}_m)} \delta(\mathbf{z}_1 - \hat{\mathbf{z}}_h)\,\delta(\mathbf{z}_{2:M} - \hat{\mathbf{z}}_{-h})\, d\hat{\mathbf{z}}_{1:M} \\
&= \sum_{h=1}^{M} \int q(\hat{\mathbf{z}}_{1:M}) \frac{p(\hat{\mathbf{z}}_h, \mathbf{x})/q(\hat{\mathbf{z}}_h)}{\sum_{m=1}^{M} p(\hat{\mathbf{z}}_m, \mathbf{x})/q(\hat{\mathbf{z}}_m)} \delta(\mathbf{z}_1 - \hat{\mathbf{z}}_h)\,\delta(\mathbf{z}_{2:M} - \hat{\mathbf{z}}_{-h})\, d\hat{\mathbf{z}}_{1:M} \\
&= M \int q(\hat{\mathbf{z}}_{1:M}) \frac{p(\hat{\mathbf{z}}_1, \mathbf{x})/q(\hat{\mathbf{z}}_1)}{\sum_{m=1}^{M} p(\hat{\mathbf{z}}_m, \mathbf{x})/q(\hat{\mathbf{z}}_m)} \delta(\mathbf{z}_1 - \hat{\mathbf{z}}_1)\,\delta(\mathbf{z}_{2:M} - \hat{\mathbf{z}}_{2:M})\, d\hat{\mathbf{z}}_{1:M} \\
&= M \int \frac{p(\hat{\mathbf{z}}_1, \mathbf{x})\, q(\hat{\mathbf{z}}_{2:M})}{\sum_{m=1}^{M} p(\hat{\mathbf{z}}_m, \mathbf{x})/q(\hat{\mathbf{z}}_m)} \delta(\mathbf{z}_1 - \hat{\mathbf{z}}_1)\,\delta(\mathbf{z}_{2:M} - \hat{\mathbf{z}}_{2:M})\, d\hat{\mathbf{z}}_{1:M} \\
&= M \frac{p(\mathbf{z}_1, \mathbf{x})\, q(\mathbf{z}_{2:M})}{\sum_{m=1}^{M} p(\mathbf{z}_m, \mathbf{x})/q(\mathbf{z}_m)} \\
&= \frac{p(\mathbf{z}_1, \mathbf{x})\, q(\mathbf{z}_{2:M})}{\frac{1}{M}\sum_{m=1}^{M} p(\mathbf{z}_m, \mathbf{x})/q(\mathbf{z}_m)}
\end{aligned}
$$

For the decomposition, we have, by Eq. 1 that

$$\log p_M(\mathbf{x}) = \mathop{\mathbb{E}}_{q_M(\mathbf{z}_{1:M})} \log \frac{p_M(\mathbf{z}_{1:M}, \mathbf{x})}{q_M(\mathbf{z}_{1:M})} + \text{KL}\left[q_M(\mathbf{z}_{1:M})\|p_M(\mathbf{z}_{1:M}|\mathbf{x})\right].$$

Now, by the definition of $p_M$, it's easy to see that $p(\mathbf{x}) = p_M(\mathbf{x})$.

Next, re-write the importance-weighted ELBO as

$$
\begin{aligned}
\mathop{\mathbb{E}}_{q_M(\mathbf{z}_{1:M})} \log \frac{p_M(\mathbf{z}_{1:M}, \mathbf{x})}{q_M(\mathbf{z}_{1:M})} &= \mathop{\mathbb{E}}_{q_M(\mathbf{z}_{1:M})} \log \frac{p(\mathbf{z}_1, \mathbf{x})\, q(\mathbf{z}_{2:M})}{\frac{p(\mathbf{z}_1,\mathbf{x})q(\mathbf{z}_{2:M})}{\frac{1}{M}\sum_{m=1}^{M} p(\mathbf{z}_m,x)/q(\mathbf{z}_m)}} \\
&= \mathop{\mathbb{E}}_{q_M(\mathbf{z}_{1:M})} \log \left( \frac{1}{M} \sum_{m=1}^{M} \frac{p(\mathbf{z}_m, x)}{q(\mathbf{z}_m)} \right).
\end{aligned}
$$

This gives that

$$\log p\left(\mathbf{x}\right) = \underbrace{\mathbb{E}_{q_M\left(\mathbf{z}_{1:M}\right)} \log\left(\frac{1}{M}\sum_{m=1}^{M}\frac{p\left(\mathbf{z}_m,x\right)}{q\left(\mathbf{z}_m\right)}\right)}_{\text{importance weighted ELBO}} + \mathrm{KL}\left[q_M\left(\mathbf{z}_{1:M}\right)\|p_M\left(\mathbf{z}_{1:M}|\mathbf{x}\right)\right].$$

$\square$

**Lemma 4.** $\mathbb{E}_{q_M(\mathbf{z}_1)} t(\mathbf{z}_1) = \mathbb{E}_{q(\mathbf{z}_{1:M})} \frac{\sum_{m=1}^{M}\omega(\mathbf{z}_m)\, t(\mathbf{z}_m)}{\sum_{m=1}^{M}\omega(\mathbf{z}_m)}.$

*Proof.*

$$
\begin{aligned}
\mathbb{E}_{q_M(\mathbf{z}_{1:M})} t(\mathbf{z}_1) &= \int \frac{t(\mathbf{z}_1)\, p\left(\mathbf{z}_1,\mathbf{x}\right) q\left(\mathbf{z}_{2:M}\right)}{\frac{1}{M}\sum_{m=1}^{M}p\left(\mathbf{z}_m,\mathbf{x}\right)/q\left(\mathbf{z}_m\right)} d\mathbf{z}_{1:M} \\
&= \int q\left(\mathbf{z}_{1:M}\right)\frac{t(\mathbf{z}_1)\, p\left(\mathbf{z}_1,\mathbf{x}\right)/q(\mathbf{z}_1)}{\frac{1}{M}\sum_{m=1}^{M}p\left(\mathbf{z}_m,\mathbf{x}\right)/q\left(\mathbf{z}_m\right)} d\mathbf{z}_{1:M} \\
&= \mathbb{E}_{q(\mathbf{z}_{1:M})}\frac{t(\mathbf{z}_1)\, p\left(\mathbf{z}_1,\mathbf{x}\right)/q(\mathbf{z}_1)}{\frac{1}{M}\sum_{m=1}^{M}p\left(\mathbf{z}_m,\mathbf{x}\right)/q\left(\mathbf{z}_m\right)} \\
&= \mathbb{E}_{q(\mathbf{z}_{1:M})}\frac{\frac{1}{M}\sum_{m=1}^{M}t(\mathbf{z}_m)\, p\left(\mathbf{z}_m,\mathbf{x}\right)/q(\mathbf{z}_m)}{\frac{1}{M}\sum_{m=1}^{M}p\left(\mathbf{z}_m,\mathbf{x}\right)/q\left(\mathbf{z}_m\right)} \\
&= \mathbb{E}_{q(\mathbf{z}_{1:M})}\frac{\sum_{m=1}^{M}\omega\left(\mathbf{z}_m\right)\, t(\mathbf{z}_m)}{\sum_{m=1}^{M}\omega\left(\mathbf{z}_m\right)}
\end{aligned}
$$

$\square$

**Theorem 2.** *The marginal and joint divergences relevant to IWVI are related by*

$$\mathrm{KL}\left[q_M(\mathbf{z}_{1:M})\|p_M(\mathbf{z}_{1:M}|\mathbf{x})\right] = \mathrm{KL}\left[q_M(\mathbf{z}_1)\|p(\mathbf{z}_1|\mathbf{x})\right] + \mathrm{KL}\left[q_M(\mathbf{z}_{2:M}|\mathbf{z}_1)\|q(\mathbf{z}_{2:M})\right].$$

*As a consequence, the gap in the first inequality of Eq 8 is exactly* $\mathrm{KL}\left[q_M(\mathbf{z}_1)\|p(\mathbf{z}_1|\mathbf{x})\right]$ *and the gap in the second inequality is exactly* $\mathrm{KL}\left[q_M(\mathbf{z}_{2:M}|\mathbf{z}_1)\|q(\mathbf{z}_{2:M})\right]$.

*Proof.*

$$
\begin{aligned}
\mathrm{KL}\left[q_M(\mathbf{z}_{1:M})\|p_M(\mathbf{z}_{1:M}|\mathbf{x})\right] &= \mathrm{KL}\left[q_M(\mathbf{z}_1)\|p_M(\mathbf{z}_1|\mathbf{x})\right] + \mathrm{KL}\left[q_M(\mathbf{z}_{2:M}|\mathbf{z}_1)\|p_M(\mathbf{z}_{2:M}|\mathbf{z}_1,\mathbf{x})\right] \\
&\quad\text{by the chain rule of KL-divergence} \\
&= \mathrm{KL}\left[q_M(\mathbf{z}_1)\|p(\mathbf{z}_1|\mathbf{x})\right] + \mathrm{KL}\left[q_M(\mathbf{z}_{2:M}|\mathbf{z}_1)\|q(\mathbf{z}_{2:M})\right] \\
&\quad\text{since } p_M(\mathbf{z}_1|\mathbf{x}) = p(\mathbf{z}_1|\mathbf{x}) \text{ and } p_M(\mathbf{z}_{2:M}|\mathbf{z}_1,\mathbf{x}) = q(\mathbf{z}_{2:M}).
\end{aligned}
$$

The KL-divergences can be identified with the gaps in the inequalities in Eq. 8 through the application of Eq. 1 to give that

$$\log p(\mathbf{x}) - \mathrm{ELBO}\left[q_M(\mathbf{z}_1)\|p(\mathbf{z}_1,\mathbf{x})\right] = \mathrm{KL}\left[q_M(\mathbf{z}_1)\|p_M(\mathbf{z}_1|\mathbf{x})\right]$$

which establishes the looseness of the first inequality. Then, Thm. 1 gives that

$$\log p(\mathbf{x}) - \mathrm{IW\text{-}ELBO}_M\left[q(\mathbf{z})\|p(\mathbf{z},\mathbf{x})\right] = \mathrm{KL}\left[q_M(\mathbf{z}_{1:M})\|p_M(\mathbf{z}_{1:M}|\mathbf{x})\right].$$

The difference of the previous two equations gives that the looseness of the second inequality is

$$
\begin{aligned}
\mathrm{ELBO}\left[q_M(\mathbf{z}_1)\|p(\mathbf{z}_1,\mathbf{x})\right] - \mathrm{IW\text{-}ELBO}_M\left[q(\mathbf{z})\|p(\mathbf{z},\mathbf{x})\right] &= \mathrm{KL}\left[q_M(\mathbf{z}_{1:M})\|p_M(\mathbf{z}_{1:M}|\mathbf{x})\right] \\
&\quad - \mathrm{KL}\left[q_M(\mathbf{z}_1)\|p_M(\mathbf{z}_1|\mathbf{x})\right] \\
&= \mathrm{KL}\left[q_M(\mathbf{z}_{2:M}|\mathbf{z}_1)\|q(\mathbf{z}_{2:M})\right].
\end{aligned}
$$

$\square$

### A.3 Asymptotics

**Theorem 3.** *For large $M$, the looseness of the IW-ELBO is given by the variance of $R$. Formally, if there exists some $\alpha > 0$ such that $\mathbb{E}\,|R - p(\mathbf{x})|^{2+\alpha} < \infty$ and $\limsup_{M \to \infty} \mathbb{E}[1/R_M] < \infty$, then*

$$\lim_{M \to \infty} M \Big( \underbrace{\log p(\mathbf{x}) - \text{IW-ELBO}_M \left[ q(\mathbf{z}) \| p(\mathbf{z}, \mathbf{x}) \right]}_{\text{KL}[q_M \| p_M]} \Big) = \frac{\mathbb{V}[R]}{2p(\mathbf{x})^2}.$$

We first give more context for this theorem, and then its proof. Since $\text{IW-ELBO}_M\,[p\|q] = \mathbb{E}\log(R_M)$ where $\sqrt{M}(R_M - p(\mathbf{x}))$ converges in distribution to a Gaussian distribution, the result is *nearly* a straightforward application of the "delta method for moments" (e.g. [4, Chapter 5.3.1]). The key difficulty is that the derivatives of $\log(r)$ are unbounded at $r = 0$; bounded derivatives are typically required to establish convergence rates.

The assumption that $\limsup_{M \to \infty} \mathbb{E}[1/R_M] < \infty$ warrants further discussion. One (rather strong) assumption that implies this[1] would be that $\mathbb{E}\,1/R < \infty$. However, this is not necessary. For example, if $R$ were uniform on the $[0,1]$ interval, then $\mathbb{E}\,1/R$ does not exist, yet $\mathbb{E}\,1/R_M$ does for any $M \geq 2$. It can be shown[2] that if $M \geq M_0$ and $\mathbb{E}\,1/R_{M_0} < \infty$ then $\mathbb{E}\,1/R_M \leq \mathbb{E}\,1/R_{M_0}$. Thus, assuming only that there is some finite $M$ such that $\mathbb{E}\,1/R_M < \infty$ is sufficient for the $\limsup$ condition.

Both Maddison et al. [13, Prop. 1] and Rainforth et al. [18, Eq. 7] give related results that control the rate of convergence. It can be shown that Proposition 1 of Maddison et al. implies the conclusion of Theorem 3 if $\mathbb{E}\left[(R - p(\mathbf{x}))^6\right] < \infty$. Their Proposition 1, specialized to our notation and setting, is:

**Proposition 1** ([13]). *If $g(M) = \mathbb{E}\left[(R_M - p(\mathbf{x}))^6\right] < \infty$ and $\limsup_{M \to \infty} \mathbb{E}[1/R_M] < \infty$, then*

$$\log p(\mathbf{x}) - \mathbb{E}\log R_M = \frac{\mathbb{V}[R_M]}{2p(\mathbf{x})^2} + O(\sqrt{g(M)}).$$

In order to imply the conclusion of Theorem 3, it is necessary to bound the final term. To do this, we can use the following lemma, which is a consequence of the Marcinkiewicz–Zygmund inequality [14] and provides an asymptotic bound on the higher moments of a sample mean. We will also use this lemma in our proof of Theorem 3 below.

**Lemma 5** (Bounds on sample moments). *Let $U_1, \ldots, U_M$ be i.i.d random variables with $\mathbb{E}[U_i] = 0$ and let $\bar{U}_M = \frac{1}{M}\sum_{i=1}^{M} U_i$. Then, for each $s \geq 2$ there is a constant $B_s > 0$ such that*

$$\mathbb{E}\left|\bar{U}_M\right|^s \leq B_s M^{-s/2}\,\mathbb{E}\left|U_1\right|^s.$$

We now show that if the assumptions of Prop. 1 are true, this lemma can be used to bound $g(M)$ and therefore imply the conclusion of Theorem 3. If $\mathbb{E}\left|R - p(\mathbf{x})\right|^6 < \infty$ then $g(M) = \mathbb{E}\left|R_M - p(\mathbf{x})\right|^6 \leq B_6 M^{-3}\,\mathbb{E}\left|R - p(\mathbf{x})\right|^6 \in O(M^{-3})$ and $\sqrt{g(M)} \in O(M^{-3/2})$. Then, since $\mathbb{V}[R_M] = \mathbb{V}[R]/M$, we can multiply by $M$ in both sides of Prop. 1 to get

$$M(\log p(\mathbf{x}) - \mathbb{E}\log R_M) = \frac{\mathbb{V}[R]}{2p(\mathbf{x})^2} + O(M^{-1/2}),$$

which goes to $\frac{\mathbb{V}[R]}{2p(\mathbf{x})^2}$ as $M \to \infty$, as desired.

*Proof of Theorem 3.* Our proof will follow the same high-level structure as the proof of Prop. 1 from Maddison et al. [13], but we will more tightly bound the Taylor remainder term that appears below.

$$\left(\frac{1}{M}\sum_{m=1}^{M} r_m\right)^{-1} = \left(\mathbb{E}_\sigma \frac{1}{M_0}\sum_{m=1}^{M_0} r_{\sigma(m)}\right)^{-1} \leq \mathbb{E}_\sigma \left(\frac{1}{M_0}\sum_{m=1}^{M_0} r_{\sigma(m)}\right)^{-1}.$$

Since $R_M$ is a mean of i.i.d. variables, the permutation vanishes under expectations and so $\mathbb{E}\,1/R_M \leq \mathbb{E}\,1/R_{M_0}$.

Let $\theta = p(\mathbf{x}) = \mathbb{E}\, R$ and $\sigma^2 = \mathbb{V}[R]$. For any $r > 0$, let $\Delta = \Delta(r) = \frac{r-\theta}{\theta} = \frac{r}{\theta} - 1$. Then $\log\theta - \log r = -\log(1 + \Delta)$. Since $r > 0$, we only need to consider $-1 < \Delta < \infty$.

Consider the second-order Taylor expansion of $\log(1 + \Delta)$:

$$\log(1 + \Delta) = \Delta - \frac{1}{2}\Delta^2 + \int_0^\Delta \frac{x^2}{1+x}dx$$

Now, let $\Delta_M = \Delta(R_M)$. Then, since $\mathbb{E}[\Delta_M] = 0$ and $\mathbb{E}[\Delta_M^2] = \frac{1}{\theta^2}\frac{\sigma^2}{M}$,

$$\mathbb{E}(\log\theta - \log R_M) = -\mathbb{E}\log(1 + \Delta_M) = \frac{1}{2}\mathbb{E}\,\Delta_M^2 - \mathbb{E}\int_0^{\Delta_M} \frac{x^2}{1+x}dx$$
$$= \frac{\sigma^2/M}{2\theta^2} - \mathbb{E}\int_0^{\Delta_M} \frac{x^2}{1+x}dx$$

Moving $M$ and taking the limit, this is

$$\lim_{M\to\infty} M(\log\theta - \mathbb{E}\log R_M) = \frac{\sigma^2}{2\theta^2} - \lim_{M\to\infty} M\,\mathbb{E}\int_0^{\Delta_M} \frac{x^2}{1+x}dx.$$

Our desired result holds if and only if $\lim_{M\to\infty}\left|M\,\mathbb{E}\int_0^{\Delta_M}\frac{x^2}{1+x}dx\right| = 0$. Lemma 7 (proven in Section A.3.1 below) bounds the absolute value of this integral for fixed $\Delta$. Choosing $\Delta = \Delta_M$, multiplying by $M$ and taking the expectation of both sides of Lemma 7 is equivalent to the statement that, for any $\epsilon > 0$, $0 < \alpha \leq 1$:

$$M\,\mathbb{E}\left|\int_0^{\Delta_M}\frac{x^2}{1+x}dx\right| \leq M\,\mathbb{E}\left[C_\epsilon\left|\frac{1}{1+\Delta_M}\right|^{\frac{\epsilon}{1+\epsilon}}|\Delta_M|^{\frac{2+3\epsilon}{1+\epsilon}}\right] + MD_\alpha\,\mathbb{E}\,|\Delta_M|^{2+\alpha}. \qquad (16)$$

Let $\alpha$ be as given in the conditions of the theorem, so that $\mathbb{E}\,|R - \theta|^{2+\alpha} < \infty$. Assume without loss of generality that $\alpha \leq 1$; this is justified because $\mathbb{E}\,|R - \theta|^{2+\alpha} < \infty \implies E|R - \theta|^{2+\alpha'} < \infty$ for all $0 \leq \alpha' \leq \alpha$. We will show that both terms on the right-hand side of Eq. (16) have a limit of zero as $M \to \infty$ for suitable $\epsilon$. For the second term, let $s = 2 + \alpha$. Then by Lemma 5,

$$\mathbb{E}\,|\Delta_M|^{2+\alpha} = \mathbb{E}\,|\Delta_M|^s = \theta^{-s}\,\mathbb{E}\,|R_M - \theta|^s \leq \theta^{-s}B_s M^{-s/2}\,\mathbb{E}\,|R - \theta|^s. \qquad (17)$$

Since $s/2 > 1$ and $\mathbb{E}\,|R - \theta|^s < \infty$, this implies that the $D_\alpha\,\mathbb{E}\,|\Delta_M|^{2+\alpha}$ is $o(M^{-1})$ and so the limit of the second term on the right of Eq. 16 is zero.

For the first term on the right-hand side of Eq. (16), apply Holder's inequality with $p = \frac{1+\epsilon}{\epsilon}$ and $q = 1 + \epsilon$, to get that

$$M\,\mathbb{E}\left[C_\epsilon\left|\frac{1}{1+\Delta_M}\right|^{\frac{\epsilon}{1+\epsilon}}|\Delta_M|^{\frac{2+3\epsilon}{1+\epsilon}}\right] \leq MC_\epsilon\left(\mathbb{E}\left|\frac{1}{1+\Delta_M}\right|\right)^{\frac{\epsilon}{1+\epsilon}}\left(\mathbb{E}\,|\Delta_M|^{2+3\epsilon}\right)^{\frac{1}{1+\epsilon}}.$$

Now, use the fact that $\limsup(a_M b_M) \leq \limsup a_M \limsup b_M$ to get that

$$\limsup_{M\to\infty} M\,\mathbb{E}\left[C_\epsilon\left|\frac{1}{1+\Delta_M}\right|^{\frac{\epsilon}{1+\epsilon}}|\Delta_M|^{\frac{2+3\epsilon}{1+\epsilon}}\right]$$
$$\leq C_\epsilon \limsup_{M\to\infty}\left(\mathbb{E}\left|\frac{1}{1+\Delta_M}\right|\right)^{\frac{\epsilon}{1+\epsilon}} \limsup_{M\to\infty} M\left(\mathbb{E}\,|\Delta_M|^{2+3\epsilon}\right)^{\frac{1}{1+\epsilon}} \qquad (18)$$

We will now show that the first limit on the right of Eq. 18 is finite, while the second is zero. For the first limit, our assumption that $\limsup_{M\to\infty}\mathbb{E}\frac{1}{R_M} < \infty$, means that for sufficiently large $M$, $\mathbb{E}\frac{1}{R_M}$ is bounded by a constant. Thus, we have that regardless of $\epsilon$, the first limit of

$$\limsup_{M \to \infty} \left( \mathbb{E} \left| \frac{1}{1 + \Delta_M} \right| \right)^{\frac{\epsilon}{1+\epsilon}} = \limsup_{M \to \infty} \left( \mathbb{E} \left| \frac{\theta}{R_M} \right| \right)^{\frac{\epsilon}{1+\epsilon}}$$

is bounded by a constant.

Now, consider the second limit on the right of Eq. 18. Let $\epsilon = \alpha/3$ and $s' = \frac{2+\alpha}{1+\epsilon} > 2$. Then, using the bound we already established above in Eq. 17 we have that

$$\left( \mathbb{E} \left| \Delta_M \right|^{2+3\epsilon} \right)^{\frac{1}{1+\epsilon}} = \left( \mathbb{E} \left| \Delta_M \right|^{2+\alpha} \right)^{\frac{1}{1+\epsilon}} \leq \theta^{-s'} B_s^{\frac{1}{1+\epsilon}} M^{-s'/2} \left( \mathbb{E} \left| R - \theta \right|^s \right)^{\frac{1}{1+\epsilon}}.$$

Since $s' > 2$ and $\mathbb{E} \left| R - \theta \right|^s < \infty$, this proves that the second limit in Eq. (18) is zero. Since we already showed that the first limit on the right of Eq. 18 is finite we have that the limit of the first term on the right of Eq. (16) is zero, completing the proof.

$\square$

### A.3.1 Proofs of Lemmas

**Lemma 6** (Bounds on sample moments). *Let $U_1, \ldots, U_M$ be i.i.d random variables with $\mathbb{E}[U_i] = 0$ and let $\bar{U}_M = \frac{1}{M} \sum_{i=1}^M U_i$. Then, for each $s \geq 2$ there is a constant $B_s > 0$ such that*

$$\mathbb{E} \left| \bar{U}_M \right|^s \leq B_s M^{-s/2} \mathbb{E} \left| U_1 \right|^s.$$

*Proof.* The lemma is proved for the case when $s$ is an integer in [4, Lemma 5.3.1]. Our proof for real $s$ follows [20]. The Marcinkiewicz–Zygmund inequality [14] states that, under the same conditions, for any $s \geq 1$ there exists $B_s > 0$ such that

$$\mathbb{E} \left( \left| \sum_{i=1}^M U_i \right|^s \right) \leq B_s \mathbb{E} \left( \left( \sum_{i=1}^M |U_i|^2 \right)^{s/2} \right)$$

Therefore,

$$\mathbb{E} \left( \left| \frac{1}{M} \sum_{i=1}^M U_i \right|^s \right) = M^{-s} \mathbb{E} \left( \left| \sum_{i=1}^M U_i \right|^s \right)$$

$$\leq B_s M^{-s} \mathbb{E} \left( \left( \sum_{i=1}^M |U_i|^2 \right)^{s/2} \right)$$

$$= B_s M^{-s/2} \mathbb{E} \left( \left( \frac{1}{M} \sum_{i=1}^M |U_i|^2 \right)^{s/2} \right)$$

Now, since $v \mapsto v^{s/2}$ is convex for $s \geq 2$

$$\left( \frac{1}{M} \sum_{i=1}^M |U_i|^2 \right)^{s/2} \leq \frac{1}{M} \sum_{i=1}^M |U_i|^s,$$

and $\mathbb{E} \left( \frac{1}{M} \sum_{i=1}^M |U_i|^s \right) = \mathbb{E} |U_1|^s$, so we have

$$\mathbb{E} \left( \left| \frac{1}{M} \sum_{i=1}^M U_i \right|^s \right) \leq B_s M^{-s/2} \mathbb{E} |U_1|^s.$$

$\square$

**Lemma 7.** *For every $\epsilon > 0$, $0 < \alpha \leq 1$, there exists constants $C_\epsilon$, $D_\alpha$ such that, for all $\Delta > -1$,*

$$\left| \int_0^\Delta \frac{x^2}{1+x} dx \right| \leq C_\epsilon \left| \frac{1}{1+\Delta} \right|^{\frac{\epsilon}{1+\epsilon}} |\Delta|^{\frac{2+3\epsilon}{1+\epsilon}} + D_\alpha |\Delta|^{2+\alpha}.$$

*Proof.* We will treat positive and negative $\Delta$ separately, and show that:

1. If $-1 < \Delta < 0$, then for every $\epsilon > 0$, there exists $C_\epsilon > 0$ such that

$$\left| \int_0^\Delta \frac{x^2}{1+x} dx \right| \leq C_\epsilon \left| \frac{1}{1+\Delta} \right|^{\frac{\epsilon}{1+\epsilon}} |\Delta|^{\frac{2+3\epsilon}{1+\epsilon}}. \tag{19}$$

2. If $\Delta \geq 0$, then for every $0 < \alpha \leq 1$,

$$\left| \int_0^\Delta \frac{x^2}{1+x} dx \right| \leq \underbrace{\frac{1}{2+\alpha}}_{D_\alpha} \Delta^{2+\alpha}. \tag{20}$$

Put together, these imply that for all $\Delta > -1$, the quantity $\left| \int_0^\Delta \frac{x^2}{1+x} dx \right|$ is no more than the maximum of the upper bounds in Eqs. (19) and (20). Since these are both non-negative, it is also no more than their sum, which will prove the lemma.

We now prove the bound in Eq. (19). For $-1 < \Delta < 0$, substitute $u = -x$ to obtain an integral with non-negative integrand and integration limits:

$$\int_0^\Delta \frac{x^2}{1+x} dx = - \int_0^{-\Delta} \frac{u^2}{1-u} du < 0$$

Therefore:

$$\left| \int_0^\Delta \frac{x^2}{1+x} dx \right| = \int_0^{-\Delta} \frac{u^2}{1-u} du$$

Now apply Holder's inequality with $p, q > 1$ such that $\frac{1}{p} + \frac{1}{q} = 1$:

$$\int_0^{-\Delta} \frac{u^2}{1-u} du \leq \left( \int_0^{-\Delta} \frac{1}{(1-u)^p} du \right)^{1/p} \cdot \left( \int_0^{-\Delta} u^{2q} du \right)^{1/q}$$

$$= \left( \frac{1}{p-1} \frac{1-(1+\Delta)^{p-1}}{(1+\Delta)^{p-1}} \right)^{1/p} \cdot \left( \frac{1}{2q+1} (-\Delta)^{2q+1} \right)^{1/q}$$

$$= C_{p,q} \cdot \left( \frac{1-(1+\Delta)^{p-1}}{(1+\Delta)^{p-1}} \right)^{1/p} \cdot \left( |\Delta|^{2q+1} \right)^{1/q}$$

$$\leq C_{p,q} \cdot \left( \frac{1}{(1+\Delta)^{p-1}} \right)^{1/p} \cdot \left( |\Delta|^{2q+1} \right)^{1/q}$$

$$= C_{p,q} \cdot \left| \frac{1}{1+\Delta} \right|^{\frac{p-1}{p}} \cdot |\Delta|^{\frac{2q+1}{q}}$$

In the fourth line, we used the fact that $0 < (1+\Delta)^{p-1} < 1$. Now set $p = 1 + \epsilon$, $q = \frac{1+\epsilon}{\epsilon}$ and $C_\epsilon = C_{p,q}$, and we obtain Eq. (19).

We now prove the upper bound of Eq. (20). For $\Delta \geq 0$, the integrand is non-negative and:

$$\left| \int_0^\Delta \frac{x^2}{1+x} dx \right| = \int_0^\Delta \frac{x^2}{1+x} dx.$$

Let $f(\Delta) = \int_0^\Delta \frac{x^2}{1+x} dx$ and $g(\Delta) = \frac{1}{2+\alpha} x^{2+\alpha}$. Then $f(0) = g(0) = 0$, and we claim that $f'(\Delta) \leq g'(\Delta)$ for all $\Delta \geq 0$, which together imply $f(\Delta) \leq g(\Delta)$ for all $\Delta \geq 0$.

To see that $f'(\Delta) \leq g'(\Delta)$, observe that:

$$\frac{g'(\Delta)}{f'(\Delta)} = \frac{\Delta^{1+\alpha}}{\frac{\Delta^2}{1+\Delta}} = \frac{\Delta^{1+\alpha}(1+\Delta)}{\Delta^2} = \frac{\Delta^{1+\alpha} + \Delta^{2+\alpha}}{\Delta^2} = \Delta^{\alpha-1} + \Delta^\alpha$$

Both terms on the right-hand side are nonnegative. Recall that $\alpha \in (0, 1]$. If $\Delta \in [0, 1]$ then $\Delta^{\alpha-1} \geq 1$. If $\Delta \geq 1$, then $\Delta^\alpha \geq 1$. Therefore, the sum is at least one for all $\Delta \geq 0$. □

### A.3.2 Relationship of Decompositions

This section discusses the relationship of our decomposition to that of Le et al. [12, Claim 1].

We first state their Claim 1 in our notation. Define $q_M^{IS}(\mathbf{z}_{1:M}) = \prod_{m=1}^M q(\mathbf{z}_m)$ and define

$$p_M^{IS}(\mathbf{z}_{1:M}, \mathbf{x}) = q_M^{IS}(\mathbf{z}_{1:M}) \frac{1}{M} \sum_{m=1}^M \frac{p(\mathbf{z}_m, \mathbf{x})}{q(\mathbf{z}_m)} = \frac{1}{M} \sum_{m=1}^M p(\mathbf{z}_m, \mathbf{x}) \prod_{m' \neq m} q(\mathbf{z}_{m'}).$$

By construction, the ratio of these two distributions is

$$\frac{p_M^{IS}(\mathbf{z}_{1:M}, \mathbf{x})}{q_M^{IS}(\mathbf{z}_{1:M})} = \frac{1}{M} \sum_{m=1}^M \frac{p(\mathbf{z}_m, \mathbf{x})}{q(\mathbf{z}_m)},$$

and $p_M^{IS}(\mathbf{x}) = p(\mathbf{x})$ and so applying the standard ELBO decomposition (Eq. 1) to $p_M^{IS}$ and $q_M^{IS}$ gives that

$$\log p(\mathbf{x}) = \text{IW-ELBO}_M \left[ q(\mathbf{z}) \| p(\mathbf{z}, \mathbf{x}) \right] + KL[q_M^{IS}(\mathbf{z}_{1:M}) \| p_M^{IS}(\mathbf{z}_{1:M} \mid \mathbf{x})].$$

This is superficially similar to our result because it shows that maximizing the IW-ELBO minimizes the KL-divergence between two augmented distributions. However, it is fundamentally different and does not inform probabilistic inference. In particular, note that the marginals of these two distributions are

$$p_M^{IS}(\mathbf{z}_1 \mid \mathbf{x}) = \frac{1}{M} p(\mathbf{z}_1 \mid \mathbf{x}) + \frac{M-1}{M} q(\mathbf{z}_1),$$
$$q_M^{IS}(\mathbf{z}_1) = q(\mathbf{z}_1).$$

This pair of distributions holds $q_M^{IS}$ "fixed" to be an independent sample of size $M$ from $q$, and changes $p_M^{IS}$ so that its marginals approach those of $q_M^{IS}$ as $M \to \infty$. The distribution one can actually sample from, $q_M^{IS}$, does not approach the desired target.

Contrast this with our approach, where we hold the marginal of $p_M$ fixed so that $p_M(\mathbf{z}_1 \mid \mathbf{x}) = p(\mathbf{z}_1 \mid \mathbf{x})$, and augment $q$ so that $q_M(\mathbf{z}_1)$ gets closer and closer to $p_M(\mathbf{z}_1 \mid \mathbf{x})$ as $M$ increases. Further, since $q_M(\mathbf{z}_1)$ is the distribution resulting from self-normalized importance sampling, it is available for use in a range of inference tasks.

## Footnotes

[1]To see this, observe that since $1/r$ is convex over $r > 0$, Jensen's inequality gives that $\left(\frac{1}{M}\sum_{m=1}^{M} r_m\right)^{-1} \leq \frac{1}{M}\sum_{m=1}^{M} r_m^{-1}$ and so $\mathbb{E}\,1/R_M \leq \mathbb{E}\,1/R$.

[2]Define $\sigma$ to be a uniformly random over all permutations of $1, \ldots, M$. Then, Jensen's inequality gives that