[Reviews · NeurIPS 2018]

Reviewer 1



== After author feedback == The generative process does indeed appear in the VSMC paper, eq. (4) in that paper discussed the joint distribution of all random variables generated. The exact expression for the distribution qM in your notation can be found in the unnumbered equation between equations (13) and (14) in the supplementary material of the VSMC paper (let T=1 and b_1=1 and the result follows). The two results are equivalent in the sense that KL(qIS||pIS) = KL(qM||pM) in their respective notations, and thus not in my opinion fundamentally different. The result was included in the original AESMC arXiv version that was published 1 year ago. However, they do illustrate two different perspectives that bridges the gap between VSMC and AESMC in a very interesting way! Especially in Thm. 2 as I mentioned in my original review. Regarding test integrals, Thm 1. relates the joint qM(z1:M) whereas you propose in eq. (9) that it is actually the marginal qM(z1) that is of interest (which is the same as the VSMC generative process of their Alg. 1). This is exactly why I believe Thm. 2 is a highly relevant and novel result that should be the focus. Theorem 3 is contained in the result from the FIVO paper, note the difference in the definition of Var(R) (in your notation) and Var(\hat pN) (in their notation). Because \hat pN is an average of iid rvs (in the IS special case), the rate follows from that Var(\hat pN)= Var(\sum_i w^i / N) = Var(w)/N which is equal to Var(R)/M in your notation. I believe the paper does contain interesting results that are novel and useful to the community, but to accurately reflect the above points would require major revision to sections 3 and 4. Since NIPS does not allow for major revision I will have to recommend reject, but would strongly encourage the authors to revise and resubmit! == Original review == The paper studies importance sampling and its use in constructing variational inference objectives. Furthermore, the authors propose to consider use of elliptical distributions for more efficient proposal distributions. The authors provide some interesting insights in the experimental sections and with the elliptical distributions. However, the generative process is not new, it is the same as in Variational SMC, and Rao-Blackwellization over the particle chosen (eq. 9 in this paper) is standard. The auxiliary variable approach (Thm. 1 in this paper) is studied in the Auto-Encoding SMC paper, see e.g. equations 10-12 in that paper. Thm 3 in this paper is studied in the Filtering Variational Objectives paper, see their proposition 1. The paper is generally well written and easy to follow. I would suggest that the authors revise to focus on Thm 2, which essentially connects Variational SMC and Auto-Encoding SMC in a very nice way, and the elliptical distributions as main contributions. Naesseth et al, Variational Sequential Monte Carlo, AISTATS 2018 Le et al, Auto-Encoding Sequential Monte Carlo, ICLR 2018 Maddison et al, Filtering Variational Objectives, NIPS 2017

Reviewer 2



Update: in light of Reviewer One's observation that Theorem 3 is a special case of Proposition 1(c) of the FVO paper, and that Theorem 1 a version -- after some manipulation -- of Claim 1 in the AESMC paper, I will downgrade my review from a 7 to a 6. ----------- The paper provides semantics for the posterior approximation that is implicit in Importance Weighted Autoencoders [4]. In particular, their key result is that by maximizing the heuristically motivated importance weighted ELBO (IW-ELBO) from [4] is equivalent to minimizing a particular KL divergence (Theorems 1 and 2). This KL divergence is not directly between an approximation and the target posterior p(z | x), but contains an additional term that quantifies the effect of importance sampling. However, this insight motivates the use of the optimizer of the IW-ELBO to compute improved estimates of posterior expectations (equation 9). The paper is well-written and well-motivated, and provides valuable semantics to the IW-ELBO, and I recommend that it be accepted. I have some concerns about the the importance weighting procedure, however -- in particular, about whether or not the weights (denoted R) can be expected to have finite variance. To be fair, these concerns also extend to the original published work [4] to some extent, although [4] skirts the problem somewhat by only concerning itself with log(R) (appendix B of [4]), whereas this paper relies on assertions about R itself. Specifically, the claim that R_M places less mass near zero for larger M (line 101) does not seem obvious to me in cases where equation (4) has infinite variance. Similarly, the usefulness of Theorem 3 depends on R having finite variance. It seems that additional conditions would need to hold to ensure that equation (4) has finite variance -- see Owen [13], example 9.1 and related discussion. The original reference Burda et al. [4] does not seem to address this problem -- they only control the mean absolute of the E[log R_M], certainly not the variance of R_M. Note also that Burda et al. [4] Theorem 1 guarantees only that the bound does not degrade, but the authors claim that the bound in fact improves. Given the tendency of vanilla KL(q || p) to have q very small where p is large, I’d like to see this potential issue addressed more directly. (At line 165 the authors say that “the variance of R is a well-explored topic in traditional importance sampling”, which is true, but I think it is still important to address the variance of R in this particular context.) In particular, I wonder whether this method could work when you don’t allow the variational parameterization to have a full covariance matrix (that is, a non-diagonal A_w in equation 14). Using a full covariance matrix is only feasible in relatively small dimensions because the number of free parameters in A_w scales quadratically with dimension. Notably, most of the experiments are all relatively low dimensional -- the clutter and dirichlet models use no more than a 10-dimensional parameter space, and even the largest “large” logistic regression model has only up to a 500-dimensional parameter. (Notably the authors do not report computation time for the logistic model.) One metric that would be revealing would be the entropy of the normalized weight distribution. Are a relatively few data points getting most of the weight after normalization? If so, it’s an indication that the importance sampling procedure is not working well. A relatively even distribution of normalized weights throughout the optimization process would be reassuring that importance weighting is doing what the authors claim. It would be more interesting to compare the logistic regression model posterior moments to an MCMC ground truth as with the other models. Minor typos: - Typo in the definition of R in line 20 -- there should be no log. - The bibliography looks strange. What are the trailing numbers? - Line 109: Can be understood - Algorithm 1. The weight function and q should be explicit or an input. (I think for theorem 1 to hold they need to be the importance weights of equation 3, and it should be clearer that they are not general.) - Theorem 1. Please define things before you refer to them (specifically p_M) - Line 184 “y” should probably be “u” - Line 191: It would probably be better to avoid double use of the letter w for both weights and variational parameters - Line 220: gradient -> gradients - I find the notation “E-IWVI” vs “IWVI” (line 231) to be a bit strange. There doesn’t seem to be anything about the importance weighting trick that requires a Gaussian approximation. So using the unmodified “IWVI” to refer to a Gaussian q and the modified “E-IWVI” to refer to an elliptical q is confusing. Perhaps “G-IWVI” could refer to the Gaussian q, and the unmodified “IWVI” can go back to referring to the importance weighting trick irrespective of the choice of q. - Double periods in the caption of figure 5.

Reviewer 3



===Update=== As Review 1 pointed out, the main result of the paper, in particular the generative process, is similar to existing literature. I downgrade my review to 6. == Original review == This paper provides a novel framework that utilizes importance weighting strategy to control the Jenson bound in KL-variational inference. It also explains re-parametrization in elliptical distributions, making the variational inference family more flexible -- especially for capturing heavy tails.
 This paper is clearly written and was easy to follow. Overall I believe it is an outstanding paper. Concerned with the usage of importance weighting, I have following questions. 1 In VAE, a tighter bound does not necessarily lead to a better inference. So I am also curious if the main goal is make inference (of the target distribution p), what it the advantage of IW? I am not extremely convinced. 2. Changing the importance weighting sample size M from 1 to infinity will change the final optimum q from mode-seeking to tail matching, without tuning the divergence. This might motivates a better tuning of M? 3 . The quick concentration of R_M to p(x) relies on Var(R), which is essentially the same asymptotic 1/M convergence rate in Theorem 3 given clt. But this result relies on the finite variance Var(R), in which case a naive importance sampling will work given the proposal q. Finite var(R) is not trivial. What will happen if Var(R)=infinity or impractically large? For example, if the target has heavier tail than the variational family, then Var(R) is never finite with any proposal q. (Sure, other tail matching methods like chi square vi will fail in this situation, too) possible typos in the manuscript: tilte: varational <- variational line 219: pairs pairs line 219: F^{-1} v <- F_w^{-1} v 
 line 277: reference